# Designer liquid-liquid interfaces made from transient double emulsions

Greet Dockx[1], Steffen Geisel[2], David G. Moore[3], Erin Koos [1], Andre R. Studart [3] & Jan Vermant[2]

Current methods for generating liquid-liquid interfaces with either controlled composition or coverage often rely on adsorption equilibria which limits the freedom to design such multi-phase materials, in particular when different components are used. Moreover, when interfaces become densely populated, slowing down of adsorption may impose additional constraints. Up to now, it is not possible to control surface coverage and composition of droplet interfaces at will. Here, we report a generic and versatile method to create designer liquid-liquid interfaces, using transient double emulsions. We demonstrate how the surface coverage in Pickering emulsions can be controlled at will, even for dense particulate layers going up to multilayers. Moreover, composite droplet interfaces with compositional control can be generated, even with particles which would have intrinsically different or even opposite adsorption characteristics. Given its simplicity, this method offers a general approach for control of composition of liquid-liquid interfaces in a variety of multiphase systems.

[1] Department of Chemical Engineering, KU Leuven, Celestijnenlaan 200F, 3001 Leuven, Belgium. [2] Soft Materials, Department of Materials, ETH Zurich, CH-8093 Zurich, Switzerland. [3] Complex Materials, Department of Materials, ETH Zurich, CH-8093 Zurich, Switzerland. These authors contributed equally: Greet Dockx, Steffen Geisel. Correspondence and requests for materials should be addressed to J.V. (email: jan.vermant@mat.ethz.ch)

Controlling interfacial composition and concentrations are key aspects in surface and interface science and technology. This has been the basis of the successful understanding and practical exploitation of solid-gas interfaces[1] and solid–liquid interfaces[2]. However, this becomes much more challenging for liquid–liquid interfaces. Generating droplets with controlled interfacial composition and coverage on demand is important for a wide variety of disciplines, including generating protocells with a complex interfacial composition[3,4]; particle-based capsules, called colloidosomes[5,6]; droplets designed for fundamental studies of transport and the role of interfacial rheological properties[7,8]; as well as generating droplets with functional composite interfaces for photonic[9] or catalytic[10] applications. Adsorption isotherms dictate the relation between the surface excess concentration and the bulk concentration. The latter may interfere with the desired properties of the emulsion, for example bulk properties may affect the properties of a dried emulsion (barrier, wetting, and transparency). It would hence be of technological interest to have a method which only deposits material at the interface. Moreover, in particular when interfaces get dense, kinetic aspects may require long contact times in traditional emulsification methods, for example when colloidal particles are involved. It can be concluded that classical emulsification methods do not give full and independent control over interfacial composition and coverage, in particular for colloidal particles, or mixtures of particles and large macromolecular objects. Such objects are particularly suitable for introducing viscoelasticity or plasticity to the interface for imparting stability to those systems[11,12]. Microfluidics seems like an obvious alternate route, and has been very powerful for a wide range of chemistries. However, in particular when trying to incorporate solid particles or larger macromolecular objects into one of the phases it often fails. Reasons for this can be as mundane as clogging due to problems with solubility or dispersibility and wetting and the operating window of these techniques is limited.

Recent advances concerning the production of emulsions and capsules have seen two important trends. On the one hand, droplets with increasingly interfacial complexity are desired for applications, such as drug delivery[13], when designing the aforementioned protocells and protocell assemblies[3,4] or for the rational design of solid stabilized emulsions[7,8]. In the latter case, arrested coalescence can be used to create supra-colloidal structures with anisotropic shape and surface chemistry as potential building blocks for novel materials[14]. On the other hand microfluidics has emerged as a technological platform where monodisperse droplets can be produced with great control over composition and size for applications in a variety of fields, such as bioassays[15] or synthesis. Scale-up of microfluidic methods has gained momentum, even for double emulsions[16], enabling production of litre quantities per day for monodisperse emulsions. The key idea is to utilize step emulsification devices in tandem to enable re-injection of pre-formed droplets. The step emulsification takes away many of the problems of parallelisation and nozzle clogging[17]. The re-injection is simple and just requires two devices to be connected, allowing efficient production of double emulsions[18].

The production of emulsions with 'on demand' surface coverage and composition remains difficult in bulk emulsification processes, such as high pressure homogenization and sonication. First, this typically leads to polydisperse emulsion droplets and second, the control over coverage and composition of the interface is limited due to the poorly defined turbulent flow conditions imposed during emulsification. Whereas microfluidic techniques offer the possibility to produce monodisperse emulsions to create droplets with controlled surface coverage and composition. However, in such microfuidic techniques the control over

coverage and composition is constrained. For example, adsorption is typically a slow process for particle stabilized systems, compared to the residence time in the microfluidic device unless special controlled flow-driven assembly methods are used[19]. Nevertheless, high particle concentrations, which may cause fouling and clogging in the microfluidic channels are typically required. Moreover, particles need to be partially wetted by the inner phase for them to be interfacially active, for the case of Pickering emulsions and their stability, but this requirement can be incompatible with the need for colloidal stability in the outer phase. Nie et al. proposed that particles be dispersed inside the droplet[20] but this puts restraints on the type of particles used, as the wettability by the final emulsion may be at odds with the aforementioned requirement. A common feature of all the above methods is that the fluids used as dispersed (inner) and continuous (outer) phases are highly immiscible, which ensures the formation of stable emulsions.

In the present work, we develop a strategy for producing droplets with controlled surface coverage and composition, where we are able to release the constraints set by either adsorption isotherms or by slow kinetics of adsorption for single emulsions. The method is based on a transient double emulsion, where the surface active materials, for example particles, are dispersed in a middle phase[21]. We are explicitly exploiting in addition phase equilibria and slow mixing of the suspending liquids leading to a transient double emulsions state to form a single emulsion with a structured fluid–fluid interface. The middle phase can be dissolved to either the inside or outside, whereas maintaining a partially immiscible three phase system. The immiscibility can come from the underlying phase diagram or be kinetic in nature. In the present approach this middle phase is chosen such that the polarity will be intermediate between the inner and outer phases, so that the requirement of interfacial activity can be reconciled with the requirement of the dispersed or dissolved material, for example, the final wetting properties of the particles for a Pickering emulsions. The middle phase is also chosen as to slowly dissolve, condensing the middle phase onto the interface. Control over composition and coverage of the interface can be achieved through the bulk concentrations and the relative flow rates of the different phases. We will discuss how this idea can be reconciled with the experimental constraints on the specific chemistries and how the different optimal microfluidic devices can be build to interrogate processes in these systems. The proposed method may be widely applicable whenever the need for controlled surface coverage and composition for liquid–liquid interfaces arises, i.e. for foams, layered films and emulsions.

## Results

**Transient double emulsion approach.** Figure 1 demonstrates the principle of the transient double emulsion approach for the case of a Pickering emulsion, but the path followed on the phase diagram is generic for other components (proteins, polymers, and polymeric surfactants). The principle is similar to liquid–liquid extraction, but with complete dissolution of the middle phase leaving the insoluble material on the interface. The different phases that constitute the transient double emulsions are ideally chosen such that before mixing they are on opposite sides of a binode of a binary or ternary phase diagram and are hence partially immiscible. Here we discuss the case of a ternary phase diagram, where more degrees of freedom can be exploited. The well-known lever rule (equilibrium) and the mass balances dictate the final compositions (for an example see Supplementary Note 1). The compositions and phase volume ratios are chosen such that the middle phase (M) either dissolves completely into either the inner phase (blue line, top diagram) or the outer,

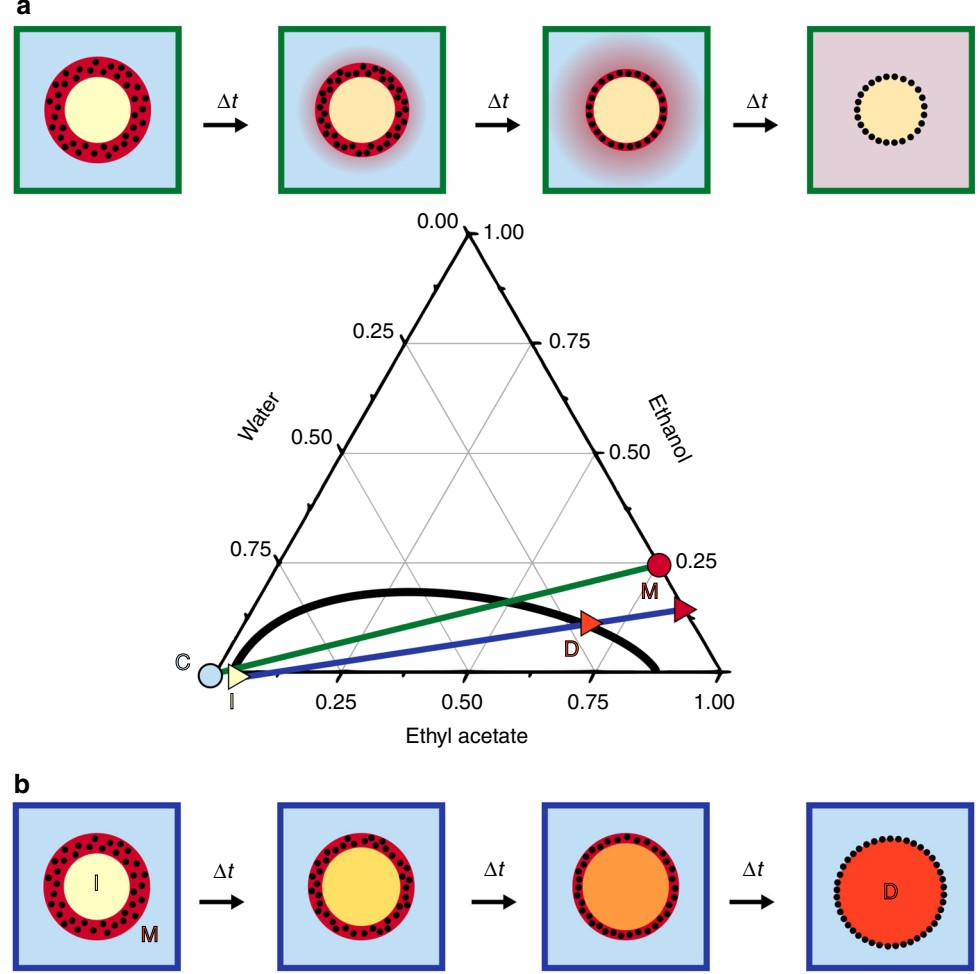

**Fig. 1** Transient double emulsion approach: Schematic representation of the transient double emulsion approach, where a middle phase (M) dissolves into the inner phase (I) or into the outer phase (O), shown, respectively, above and below the phase diagram. The particles are always suspended in the middle phase. Example phase diagram for an oil-in-water emulsion where volumes are chosen in a way that the mixing point, i.e. the composition of the final droplet (D), lies on the binodal curve. For the green schematic **a**, water is used as the outer phase (C) and an inert oil is the inner phase (not shown on the diagram)), for the blue schematic **b** an inert oil is used as the outer phase (not shown on the diagram)

continuous phase (green line, lower diagram) as shown in Fig. 1. The material that is intended to be on the interface of the final emulsion droplet first can be dissolved or dispersed in the middle or auxiliary phase, which can be chosen to have an intermediate polarity compared to the outer and inner phases. In the example of Fig. 1 the point M should be somewhere on the right side of the binode. The precise composition of this auxiliary middle phase can be chosen as to accommodate solubility or dispersion stability constraints. This alleviates many of the constraints set in current single emulsion phase methods (e.g. inside out method, or particle adsorption). In a first scenario (blue line) the middle phase lies on the other side of a binode compared to the inner droplet phase (I). After generating the double emulsions droplet, the middle phase starts to dissolve into the inner droplet phase. The compositions and volume ratios need to be chosen such that the system stays biphasic during the whole cycle, but then the middle phase disapears as to end up as a single phase emulsion. Alternatively, in the scenario as for compositions connected by the green line, we can have the middle phase dissolve into the outer phase (C), where again the lever rule and the mass balance dictate the final compositions and sizes. This scenario is in practice easier to realize. The way in which this is achieved in a microfluidic system is shown in Fig. 2a. Glass capillary devices with planar Y- or T-junction type designs are typically used. In particular the T-

junctions are well suited for generating double emulsions without too stringent demands on wettability[22,23]. The geometric design used for the microfluidic chip consists of two flow focusing geometries in series, with either two T-junctions or one Y and one T-junction as schematically depicted in Fig. 2a. In this example, a first junction brings the middle and inner phase into contact. At the second T-junction a double emulsion droplet is generated.

Specifically, Fig. 2b shows a sequence for an oil-in-water emulsion in a chip as in Fig. 2a. In the first step, an inert inner fluorinated oil phase (Fluorinert FC40, 3M company, the flow rate $Q_i = 500 \, \mu l \, h^{-1}$) is brought into contact with the future middle phase containing $12 \, mg \, ml^{-1}$ 1% ODTMS treated fluorescent silica particles (diameter = 600 nm) dispersed in a mixture of ethylacetate (75 mole%) and ethanol (25 mole%). In the second step a droplet is generated using pure water as the outer phase. The relative flow rates between the outer water phase ($Q_o = 7000 \, \mu l \, h^{-1}$) and the middle phase ($Q_m = 500 \, \mu l \, h^{-1}$) dictate the sizes of the droplet and shell and together with the phase equilibrium at 293.15 K (shown in Fig. 1) this determines both the pathway to and the composition of the final emulsion. There are hence several degrees of freedom (chemicals used, composition, flow rates, phase equilibrium, and temperature), which makes this a very flexible method. The middle phase slowly dissolves into the water as the droplets move inside the

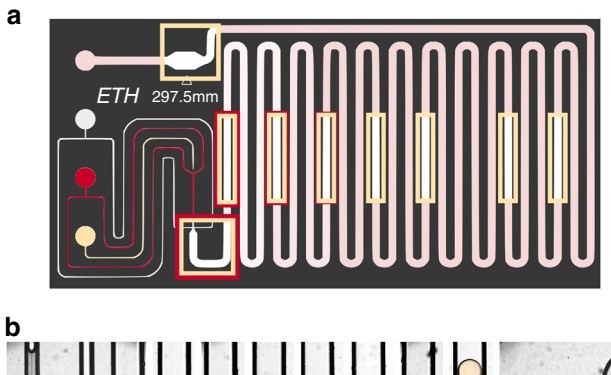

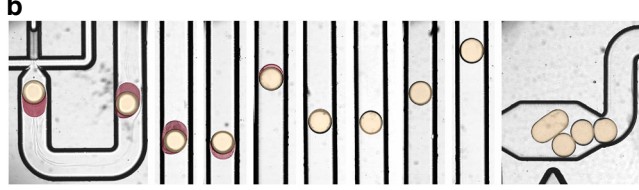

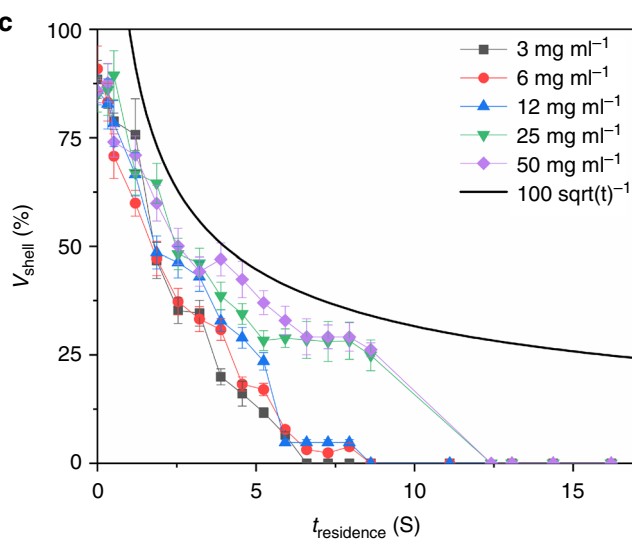

**Fig. 2** Transient double emulsions. **a** Microfluidic chip with the inlets for the outer (white), middle (red), and inner (beige) phases, indicating positions where droplets were imaged. **b** Transient droplets as imaged in different positions along the channel (indicated in **a**) for a system as in Fig. 1 (false coloured for clarity) with a particle concentration of 12 mg ml$^{-1}$. **c** Quantification of the middle droplet phase volume as a function of residence time in the channel, for different particle loadings in the middle phase. The error bars in **c** represent the standard deviation of the shell volume calculation

panel a) showing the shell (false coloured red) shrink and creating stable droplets. The specific experiment shown here was at the minimal surface coverage where we expect stability against coalescence. The coverage was homogeneous from one drop to another, but exactly at 12 mg ml$^{-1}$ we observe accidental coalescence (which is not a fully deterministic process). The coalesced droplets remained anisotropic in shape, confirming the near maximum packing of the surface coverage just before coalescence. This will be addressed in more detail below.

**Control over surface coverage**. A major advantage of the proposed transient double emulsion approach is that the surface active material is only present at the interface, which is particularly relevant for microfluidic methods and their scale-up, as risks for clogging are minimized. This is evident in Fig. 3b where a droplet, collected from a typical transient double emulsion experiment is imaged by confocal microscopy (details of this specific droplet production are given in Supplementary Note 2). The surface coverage in this case was controlled to be above maximum packing. Fluorescent particles are only found at the droplet interface and not inside or outside of the droplet, showing the concept to work. At $h = 19.1\,\mu m$ a satellite droplet is observed and even that one only has particles at the interface. However, due to the large density difference between the fluorinert and the aqueous phase, the droplet sediments to the bottom and deforms making it difficult to quantify the surface coverage precisely with this system. However, the droplets are observed to resist coalescence as the surface coverage is increased, starting at about 6 mg ml$^{-1}$ corresponding to a surface coverage of about 45%, with stable droplets being observed at concentrations of 12 mg ml$^{-1}$ and above (see an assessment of emulsion stability in Supplementary Fig. 2). Note that at concentrations well above 12 mg ml$^{-1}$ multilayers are expected.

The requirement that the middle and either inner or outer phases straddle the two sides of a binode could be construed to be a serious limitation. Therefore, we investigated whether this condition could be relaxed and a miscible system was used, where the mixing of inner and middle phases is slow enough so that the underlying concept of a transient double emulsion could be generalised to a simplified method, which works for dissolving the middle phase into the inner one. Oil-in-water emulsions, consisting of 1-butanol and n-hexadecane respectively as middle and inner phase and saturated water as continuous phase were successfully produced. Additionally, water-in-oil emulsions, consisting of pure water and an aqueous glycerol solution (20 wt%) as middle and inner phase, respectively, and decanol as continuous phase were also generated. Although, the transient flows occurring during contact and mixing of the inner and outer phase are even more complicated than those observed in the experiments of Fig. 2b, the resulting emulsions exhibit identical features and conserve the advantages of the method in both exploiting the intermediate polarity of the middle phase and the fact that particles condense onto the interface with none remaining in the inner or outer phase, at least provided mixing is slow enough.

The droplet coverage $\Phi_s$ is controlled by the flow rate of the middle phase (which controls the shell thickness), the bulk concentration of particles, and the surface area of the droplet. The droplet area is set by the flow ratios between inner and outer phase and by the ripening of the droplet, in the case where the middle and inner phase are miscible (and the outer phase is not) and a simple mass balance gives the surface coverage (see Supplementary Note 1). To be able to observe the surface coverage precisely long term visualisation was required and deformation of the droplets or gradients in particle concentration

microfluidic channel, whereas maintaining an interface at all times. Figure 2c shows the evolution of the shell volume of the double emulsion as a function of residence time by taking images at different positions in the channel for different particle concentrations between 3 and 50 mg l$^{-1}$. Maximum packing of a monolayer is expected near 12 mg ml$^{-1}$ as calculated based on the mass balance (see Supplementary Note 1). A maximum residence time of about 15 s in the microfluidic channel was required for the systems used here, at the chosen flow rates and phase ratios. At the highest particle volume fractions, the dissolution rate slows down somewhat due to the presence of the particles which hinder the diffusion. But in the ideal transient double emulsion method the three phase system is maintained up to the point where the middle phase disappears and particles are kept in the interface by the interfacial tension. Figure 2b shows a sequence of images along positions in the channel (as indicated in

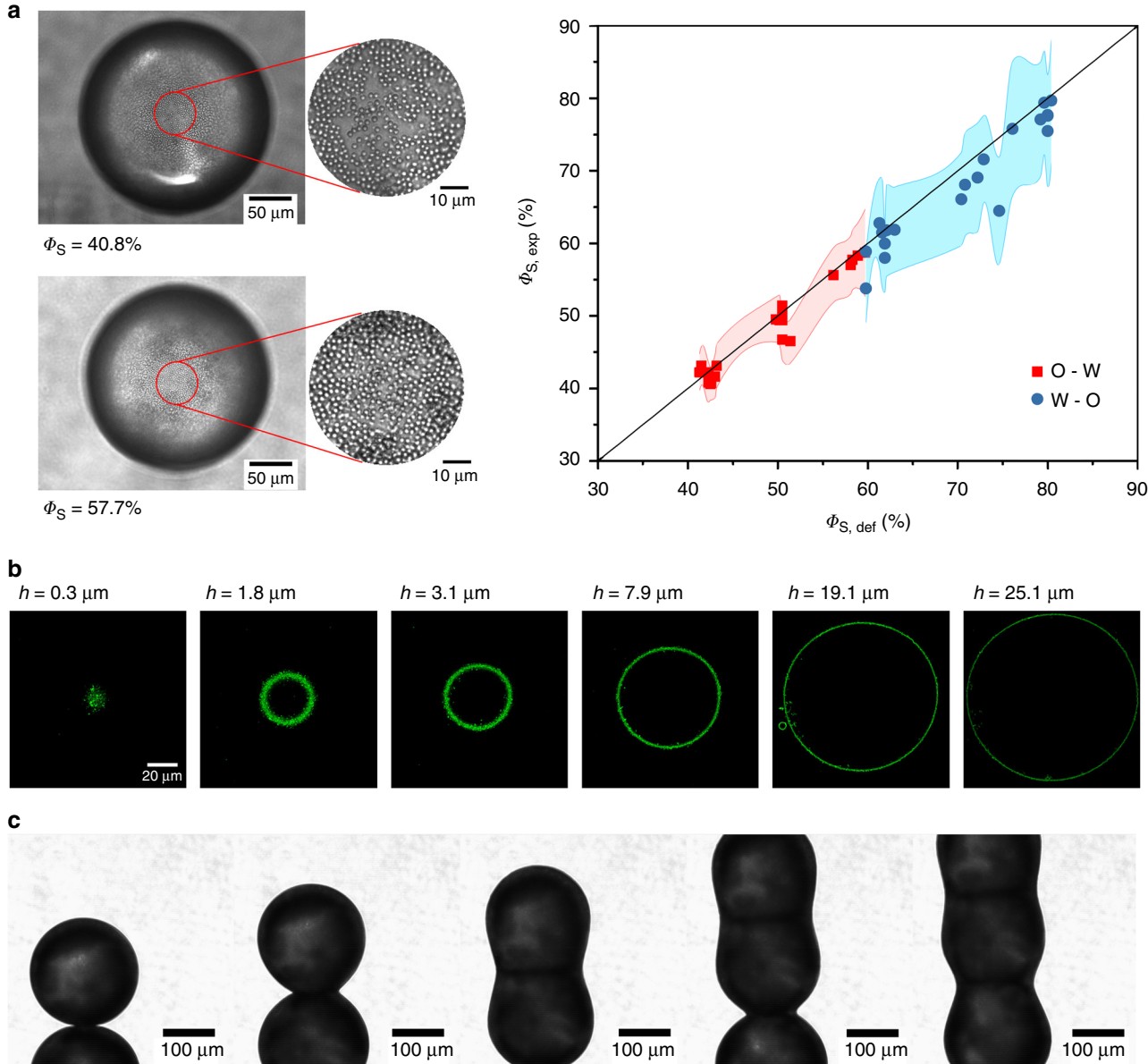

**Fig. 3** Control over surface coverage. **a** Micrographs of typical silica particle stabilized droplets at $\Phi_s = 40.8\%$ and $\Phi_s = 57.7\%$ and the comparison between the desired ($\Phi_{s,def}$) versus the experimentally obtained $\Phi_{s,exp}$ surface coverage for oil-water and water-oil systems, with the shaded areas a conservative error obtained from the image analysis as explained in the text. **b** Confocal image of a single droplet showing particles only on the interface. **c** Arrested coalescence of droplets with an initial surface coverage $\Phi_s = 85.5\%$ which due to surface jamming halt shape relaxation

due to the density of the particles had to be prevented. To this end hollow silica particles with a diameter of 1.9 µm were synthesized and functionalized with 5 wt% ODTMS for 5 h and dispersed in the middle phase consisting of pure 1-butanol (VWR Chemicals) with a particle concentration of 8.9 wt%. As an inner phase n-hexadecane (99%, Acros Organics) was used. Water was saturated with 1-butanol and employed as the continuous phase. For the microfluidic experiments the flow rates of the inner and continuous phase were kept constant at 1000 µl h⁻¹ and 15,000 µl h⁻¹, respectively. The flow rate of the middle phase was varied to vary the surface coverage. Both oil-in-water and water-in-oil emulsions could be prepared and the droplets were trapped by clamping off the in- and outlet of the microfluidic chip and imaged using optical microscopy. The number of particles within a circle on the bottom of the droplet was counted using a routine written in ImageJ. Knowing the radius of the circle, the area of the corresponding sphere segment was calculated. The error was

estimated by evaluating the sensitivity of the method by slight changes of the circle radius, due to particles lying on the edge of the circle. Figure 3a compares two droplets where the goal was to have a droplet surface coverages of 40% and 58%, respectively. The obtained surface coverages were 40.8% and 57.7%. The graph shows the desired ($\Phi_{s,def}$) versus the experimentally obtained $\Phi_{s,exp}$ surface coverage, with the shaded region a conservative error obtained from the image analysis. It can be concluded that even the simplified double emulsion templating method shows an as yet unprecedented control over surface coverage.

**Controlled arrested coalescence**. To prove this even more clearly we prepared emulsions covered with hollow silica spheres, at a surface coverage just below maximum surface coverage ($\Phi_s = 85.5\%$). These particles will act as near ideal hard sphere systems, and the surface pressure and interfacial rheology are expected to

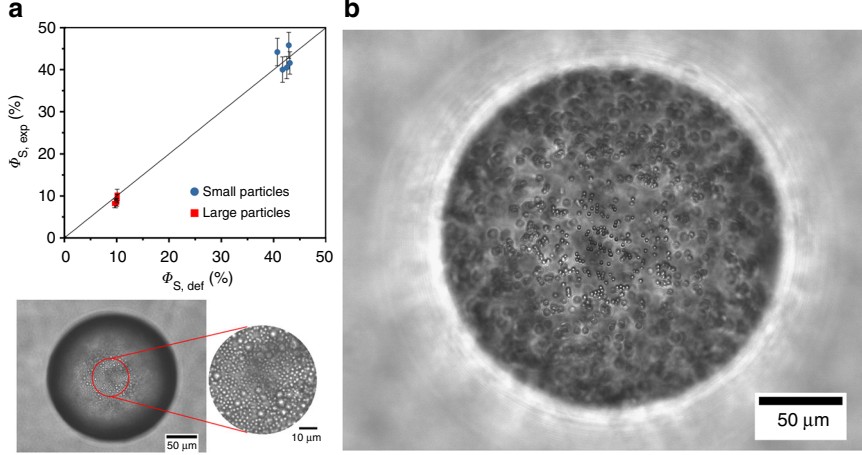

**Fig. 4** Control over composition. **a** Micrograph of emulsion droplet with a predetermined mixture of large and small silica particles, quantified using image analysis. **b** Composite emulsion droplets stabilised using a combination of hydrophobic silica spheres and hydrophylic rodlike imogolite particles, this image was taken after the miscible liquid had dissolved and a multilayer was formed

rise steeply only near maximum packing[12,24] and the divergence of the elastic moduli in shear and dilation at maximum packing (jamming) is expected to create a resistance to deformation, such as the one provided by the Laplace pressure driving force when two droplets are merged. Arrested coalescence of droplets has been suggested to have a broad impact on commercial food production and for fabricating novel anisotropic colloidal microstructures[14]. However, arrested coalescence does not occur frequently in bulk emulsification, and micro-manipulation has been proposed, which is not very scalable[25]. In the present experiments, droplets with a surface coverage of 85.5% were brought together in a diverging zone of the microfluidic chip. Figure 3c shows how the droplets merge, but coalescence is clearly arrested. During coalescence, the surface to volume ratio changes and the interface is compressed leading to a higher surface coverage. In the experiment first a snow man was created as shape relaxation is arrested by interfacial jamming. By going from two separate droplets to a snow man the surface coverage changes from 85.5 to 93.8%. When more droplets arise, highly controlled anisotropic supra-colloidal sausages are created. A movie showing the formation in real time is shown in Supplementary Movie 1. Besides its technological relevance the use of the double emulsion templating method for controlled surface coverage enables one to investigate fundamental processes in emulsions with complex structured interfaces with high resolution and precision.

**Control over surface composition**. Apart from control over coverage, the transient double emulsion methods enables unprecedented control over composition. The sole requirement for the material to end up at the interface is that the surface active material can be dispersed in the middle phase. This releases or at least simplifies many of the inherent limitations set by transport phenomena or thermodynamics. To create an interface with a predetermined composition, their bulk concentration in the middle phase needs to be controlled, in addition to the phase ratios (inner/middle/outer). Figure 4 gives two examples where this was used to create emulsions droplets stabilized using a mixture of large and small silica spheres. The ratio of large to small particles was varied and controlled. The surface coverage of small particles was set to be 50 and 10% of the surface was covered with large particles. The resulting droplets had coverage

of on average 42.4% small ones, and 8.9% large ones. It can be concluded droplets with controlled topology or chemical heterogeneity can be generated using the transient double emulsion templating approach, which could be further perfected when 2D phase behaviour on curved surfaces would be exploited[26,27]. Even droplets with composite particle layers can be achieved. This is shown, as an example for a droplet stabilized by combination of hydrophobic hollow spherical silica particles (2 wt%) and more hydrophyilic imogolite nanorods (1.9 wt%), which have been shown to be surface active[28]. It is difficult to imagine other methods where particles of such different nature can be assembled at the interface in a single step.

**Discussion**

Concerning control of the process, as with all processes for creating double emulsions wetting issues are of prime importance. Once stable transient emulsions are formed the main timescale, which needs to be controlled is the one of dissolution of the shell, relative to the length of the channel. When the timescale of sedimentation of particles is short compared to this dissolution time scale, Marangoni or other types of mixing flows occur, which lead to a homogenization of the particles on the surface in the experiments we have performed. When particle sedimentation is a significant problem it will first and foremost occur in the syringe and tubing delivering the middle phase to the microfluidic chip as flow rates are small. The main concern for the chemicals is that the middle or auxiliary phase should enable dispersion or dissolution of the material which one wants to deposit onto the interface. This middle phase will always be intermediate between the two other phases in terms of polarity, and for the surface active material to be surface active that will often be the case as well. Moreover, we have found that the condition of transient immiscibility could be relaxed to one of slow dissolution.

Concluding, exploiting phase diagrams, or slow miscibility, a transient double emulsion approach provides excellent control over composition and coverage. The method presented here is not limited to particles, but provides good control for types of emulsions stabilised by insoluble Langmuir mono- or even multilayers. For soluble components, the method may provides routes which may be faster or prevent the system from being blocked by certain kinetic pathways. The method is expected to be both of

fundamental interest and of practical importance. From a fundamental perspective, the control over surface coverage and composition enables one to tailor and investigate the interfacial rheological properties which underly emulsion stability. This should lead to engineering guidelines for the selection or development of the optimally performing interfacial stabilisers. From a practical point of view, droplets with composite surfaces with chemical or physical heterogeneities could lead to a variety of applications.

## Methods

**Microfluidic chips.** Glass chips were fabricated in the FIRST lab at ETH Zurich using wet etching techniques as described by Ofner et al.[16]. A 1.0 mm thick Borofloat 33 wafer was etched to 125 μm and diced into $15 \times 60$ mm chips. 0.7 mm inlets were drilled using a diamond-coated drill bit. The final glass microfluidic chip was obtained by sealing two symmetric glass chips so that the height of the channels is 250 μm. The sealing was controlled by visual inspection and flushing the devices with water and acetone before starting the experiments. If leakage occurred, the devices were put in the oven at 550 °C overnight with a weight on top that isotropically applies a pressure on the device of ~10 kPa.

**Experimental setup.** The device was rinsed for a few minutes with a 0.25 M aqueous NaOH solution and water after which the three fluids were pumped through the microfluidic device using syringe pumps (Harvard Apparatus) forming double emulsions. The PTFE tubing and connector were purchased from Achrom, and Dolomite Microfluidics, respectively. The formation of the double emulsion template was observed by an inverted optical microscope (Nikon Eclipse Ti) equipped with a high-speed camera (Fastcam Mini UX50, Photron). The emulsion droplets were collected in a glass vial and pipetted onto a glass coverslip, or directly on a glass coverslip (Menzel-Gläser, $24 \times 60$ mm, #1) or Petridish (MatTek Corporation, 35 mm Petridish, 20 mm Microwell, #1.5) to be able to directly observe under an optical or confocal microscope. Prior to each experiment, the chip was flushed with NaOH (1 M) in the case of oil-in-water or with silanization solution I (5% in Heptane, Sigma-Aldrich) in the case of water-in-oil emulsions.

**Particles.** Fluorescent silica particles were synthesised by a modification of the Stober method as described by van Blaaderen and Vrij[29]. The particles were hydrophobized using Octadecyltrimethoxysilane (ODTMS, 90%, Acros Organics) and butylamine (99.5%, Sigma-Aldrich). Hollow silica particles were synthesized using polystyrene particles as a Template on which a silica shell was grown following Chen et al.[30]. The particles were hydrophobized using Octadecyl-trimethoxysilane (ODTMS, 90%, Acros Organics) and butylamine (99.5%, Sigma-Aldrich). The particles were dispersed in isopropanol until no more sediments were visible. ODTMS was added together with butylamine (10 vol%, relative to ODTMS). The dispersion was stirred for several hours and subsequently centrifuged and dried at 80 °C. Imogilites were synthesized according to the protocols given in Picot et al.[28].

**Surface coverage quantification.** 3D image acquisition of the droplets were performed by fluorescent imaging using a Leica TCS SP8 inverted confocal laser scanning microscope equipped with ×20 and ×63 glycerol-immersion objectives (HC PL APO ×20/0.75 NA IMM CS2 and HC PL APO ×63/1.3 NA GLYC CORR, respectively). The dyes FITC and RhB-ITC are excited by using 488 nm and 552 nm wavelengths, respectively. The microscope is equipped with a spectral detector allowing the detected wavelength range to be adjusted in order to minimize crosstalk between the emission signals if a mixture of dyes is used. The image detection pinhole size was set to one airy disk unit while full-frame ($1024 \times 1024$ pixels) confocal images were acquired. Fast fluorescent imaging of the microfluidic experiments was performed using a commercial-based multi-beam confocal microscope, which uses structural illumination to resolve resolution issues (Visi-Tech International iSIM, UK). An inverted microscopy body (Nikon Ti-E Eclipse) is equipped with ×20 air and ×40 water-immersion objective (Plan Apo VC 0.75 NA and Apo λS 1.25 NA, Nikon, respectively) using a fibre coupled 561 nm diode with complementary dichroic mirrors and filters, imaged on an EM CCD camera (ORCA-Flash4.0 V2, Hamamatsu). The image detection pinhole size was set on 64 μm while full-frame ($1536 \times 1536$ pixels) and $1536 \times 632$ pixels frame confocal images were acquired with exposure times ranging from 2 ms to 31 ms. Bright field microscopy was also used to quantify the number of particles on a droplet. The droplets were trapped inside the microfluidic chip and imaged. The amount of particles within a circle on the bottom of the droplet was counted (using the image analysis program ImageJ). Knowing the radius of the circle, the area of the corresponding sphere segment was calculated. To have a conservative error estimate of the image analysis program, the radius of the circle was varied over a thickness of 1 particle diameter and the resulting variations in surface coverage were calculated.

## Data availability

The data that support the findings of this study are available from the corresponding author upon reasonable request.

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

## Acknowledgements

The authors acknowledge funding from the Fund for Scientific Research—Flanders and the Swiss National Science Foundation. The authors thank Kirill Feldman for the synthesis of the immogolites

## Author contributions

J.V. conceived the work. G.D. and S.G. carried out the experiments. D.M. designed and produced the microfluidic chip. S.G., G.D., E.K., A.R.S. and J.V. contributed to discussions. G.D., S.G. and J.V. wrote the manuscript.

## Additional information

**Competing interests:** The authors declare no competing interests.

