## [Peer Review File · Nature Communications]

Reviewers' comments:

Reviewer #1 (Remarks to the Author):

The manuscript presents a novel method of controlled production of emulsion droplets with adsorbed colloids. A clever technique is used that controls compositional changes and allows the particles to be added via a suspension with a second liquid phase, then dissolves the middle liquid phase to create armored droplets.

Surfaces with adsorbed and arrested colloids are increasingly of interest, and have been demonstrated to have a number of possible applications. It is timely for developments to be made that allow their design, use, and robust production. This work describes such a method by combining knowledge of formulation methods with understanding of fluid flow and colloidal structures.

I expect this work to influence both academic and industrial researchers, by demonstrating an example of how to design processes that allow engineering of advanced mesostructured materials. The work should be published but first should be clarified in some minor respects to answer some of the questions and comments below:

1. This sentence does not make sense to me, is there a word missing?

"The surface concentrations are then determined by bulk concentrations (or activities) and the latter may not necessarily be chosen free..."

2. The authors don't mention density effects of the adsorbed particles. For systems where coverage is on the order of 50%, is the time scale of sedimentation of the particles short compared to the dissolution of the middle phase? Is there a set of conditions where dense particles might segregate to one hemisphere of the droplet and need to be moved back across the surface by dissolution of middle phase?

3. This work offers an elegant approach to droplet and capsule design and production with unprecedented control. However there are a number of time scales characteristic of the process that could be used to enable quantitative design. One example is the rate of colloid relaxation and jamming during middle phase dissolution, which might be affected by the middle phase viscosity. Building on (2), another time scale is that of sedimentation of dense particles. Can some guidance be offered as to characteristic times relevant to these processes? As fluids with different viscosities could be used, as well as particles with significant density variations, it would be good to know the limits of the process.

Reviewer #2 (Remarks to the Author):

This paper describes an interesting method of controlling the density and composition of interfacial particles of particle-stabilized emulsions. By using transient double emulsion droplets with particles dispersed in the middle phase and letting the middle phase disappear through dissolution into the inner or outer phases of the double emulsion, Pickering emulsion with desired particle coverage and composition can be produced. The work is done competently and demonstrates that the proposed method can lead to Pickering emulsions of controlled compositions and coverage. It is shown that highly covered emulsion droplets can undergo arrested coalescence to produce anisotropic shapes. Overall, the work is done with care and thoroughness. I am a bit hesitant in recommending acceptance of this work. The reason is I am not sure how easily this method can be implemented by those in the field. While it is true that microfluidic methods to produce emulsions have become quite popular in the field, it is still limited to a small number of research groups that

have the specific facilities and skill sets. How scalable would this technique be? It is true that recent advances in scale-up of microfluidics have been made but they are mostly limited to single emulsion scale-up. Using double emulsions to control interfacial compositions also have been demonstrated in the fabrication of liposomes (Weitz et al.). Also, while the work presented is performed thoroughly, the manuscript itself does not present detailed characterization of the monolayer (other than its composition and packing density) nor does it provide some specific applications that can only be enabled by this approach. More specific examples of how these tailored particle-stabilized emulsions could be superior to the conventional Pickering emulsions would potentially make this work suitable for a Nature journal. For these reasons, I feel that the paper is likely better suited for a more area-specific journal, rather than Nature Communications. I also have a few additional questions:

1. I am a bit confused by the how the inner phase evolves over time as its composition changes from Point I to Point D in Figure 1. Is there any phase separation that is observed during the evolution of the droplet morphology as the middle phase dissolves into the inner phase?
2. Could this method be used to produce sub-micron/nano droplets? These are the droplets that are very difficult to make and control using the conventional methods of emulsification.
3. Figure 1S(g) shows that some droplets remains as single droplets whereas some undergo arrested coalescence. Isn't this showing that not all droplets have uniform coverage and there are some heterogeneity in the particle coverage as also shown in the non-negligible scatter and error range in Figure 3(a)?

Reviewer #3 (Remarks to the Author):

The paper "Designer liquid-liquid interfaces made from out-of-equilibrium double emulsions" demonstrates a new method for producing particle stabilized droplets with the following advantages: controlled surface coverage; uniform surface coverage over a population of droplets; monodisperse droplets made in microfluidics. The first two points in particular represent an outstanding challenge in the field of emulsification and formulations, that is addressed here for the first time. The proposed approach is very attractive and I have no doubt that it will be taken up by many researchers in the field.

The idea presented is simple and powerful. Droplets are formed from two immiscible phases as usual. The material that is to form the coating on the droplet surface is suspended in an auxiliary phase, which is miscible with either the continuous or the dispersed phase. A double emulsion is formed, so that the a layer of the auxiliary phase initially separates the two immiscible phases. The auxiliary phase diffuses in the phase in which it is miscible, and the suspended material is left as a coating on the droplets. The parameters that determine the surface coverage (flow rate of auxiliary phase, etc) are all controlled in practice, unlike in existing methods.

I really like the concept presented in this paper and the results shown are very compelling. My questions and comments are below.

COMMENTS

1. I have a question as to the generality of this approach, since it requires identifying an auxiliary phase with suitable miscibility, and also in which the material (colloids in this case, but could be proteins or polymers) is stable in the colloidal sense. Could the authors reassure us in this respect?
2. I would like to encourage the authors to take some time to improve the clarity of the manuscript, which is a bit difficult to follow at the moment. The first three sentences in the

abstract are obscure to a non-expert reader. The Introduction is unclear: thermodynamics, kinetics, and other fundamental concepts are mentioned superficially but the details relevant to the system of interest remain vague (e.g. what is meant with "The immiscibility can come from the underlying phase diagram or be kinetic in nature"?). The Introduction is also a bit repetitive, especially on the discussion of microfluidics approaches.

3. What are the shaded areas in the graph in Figure 3a?

MINOR POINTS

4. Non-equilibrium double emulsion (as in the title) or transient double emulsion (as in the paper)? I personally find "transient" to more clearly convey the mechanism behind this new approach

5. Figure 4b left me wondering if in this image the particles are in a monolayer at the interface, in a multi layer at the interface, or in the miscible liquid layer before it dissolves. Can this be made clearer in the caption?

Replies to the comments of the reviewers.

We thank the reviewers for their positive feedback on the manuscript. We provide a point-to-point reply to their specific comments. The reviewer's comments are given in black, font Times, and our replies in blue, font Arial. The revised sentences from the manuscript are given in italics.

Replies to the comments of reviewer 1.

The manuscript presents a novel method of controlled production of emulsion droplets with adsorbed colloids. A clever technique is used that controls compositional changes and allows the particles to be added via a suspension with a second liquid phase, then dissolves the middle liquid phase to create armored droplets.

Surfaces with adsorbed and arrested colloids are increasingly of interest, and have been demonstrated to have a number of possible applications. It is timely for developments to be made that allow their design, use, and robust production. This work describes such a method by combining knowledge of formulation methods with understanding of fluid flow and colloidal structures.

I expect this work to influence both academic and industrial researchers, by demonstrating an example of how to design processes that allow engineering of advanced mesostructured materials. The work should be published but first should be clarified in some minor respects to answer some of the questions and comments below:

1. This sentence does not make sense to me, is there a word missing?

"The surface concentrations are then determined by bulk concentrations (or activities) and the latter may not necessarily be chosen free..."

When adsorption is used to control surface concentration a certain bulk concentration will be required as dictated by the Gibbs adsorption isotherm. The bulk concentration could impact the properties of the bulk phases as well, for example when an ionic surfactant is used it changes the ionic strength of the bulk medium which could change the colloidal stability of the emulsion. Or when an emulsion is dried the compositions of the continuous phase will strongly affect the final film properties (barrier, transparency). To clarify this, we have rewritten this sentence.

"Adsorption isotherms dictate the relation between the surface excess concentration and the bulk concentration. The latter may interfere with the desired properties of the emulsion, for example bulk properties may affect the properties of a dried emulsion (barrier, wetting, transparency)."

2. The authors don't mention density effects of the adsorbed particles. For systems where coverage is on the order of 50%, is the time scale of sedimentation of the particles short compared to the dissolution of the middle phase? Is there a set of conditions where dense particles might segregate to one hemisphere of the droplet and need to be moved back across the surface by dissolution of middle phase?

We thank the reviewer for bringing this up. However, this is not an issue we have observed or struggled with. When sedimentation is a problem it will first and foremost occur in the syringe and tubing delivering the middle phase to the microfluidic chip as flow rates are small. Moreover we have observed that during dissolution of the middle phase, Marangoni or other types of mixing flows occur which will lead to a homogenization of the particles (or any other surface active moieties). We have added a clarification to the manuscript on page 16/17 (also see next comment).

3. This work offers an elegant approach to droplet and capsule design and production with unprecedented control. However there are a number of time scales characteristic of the process that could be used to enable quantitative design. One example is the rate of colloid relaxation and jamming during middle phase dissolution, which might be affected by the middle phase viscosity. Building on (2), another time scale is that of sedimentation of dense particles. Can some guidance be offered as to characteristic times relevant to these processes? As fluids with different viscosities could be used, as well as particles with significant density variations, it would be good to know the limits of the process.

Due to the occurrence of the aforementioned convection effects, it is difficult to evaluate the different timescales by simple scaling arguments. To accommodate the reviewers comment we have added a short discussion in the text on page 16/17. The two main timescales which are important are the sedimentation time scale in the tubes towards the chip, and as discussed in the original manuscript the dissolution time scale. The viscoelastic relaxation times of the interface are not that important, as the droplets are collected and time to relax can be as long as one desires. For yield stress interfaces or jammed materials, varying the rate of dissolution actually provides a unique method to vary the kinetic pathway by which these interfaces are generated. To integrate the reviewers' relevant remarks we have added comments to the text on page 16/17.

“Once stable transient emulsions are formed the main time scale which needs to be controlled is the one of dissolution of the shell, relative to the length of the channel. When the time scale of sedimentation of particles is short compared to this dissolution time scale, Marangoni or other types of mixing flows occur which lead to a homogenization of the particles on the surface in the experiments we have performed. When particle sedimentation is a significant problem it will first and foremost occur in the syringe and tubing delivering the middle phase to the microfluidic chip as flow rates are small.”

Replies to the comments of reviewer 2.

This paper describes an interesting method of controlling the density and composition of interfacial particles of particle-stabilized emulsions. By using transient double emulsion droplets with particles dispersed in the middle phase and letting the middle phase disappear through dissolution into the inner or outer phases of the double emulsion, Pickering emulsion with desired particle coverage and composition can be produced. The work is done competently and demonstrates that the proposed method can lead to Pickering emulsions of controlled compositions and coverage. It is shown that highly covered emulsion droplets can undergo arrested coalescence to produce anisotropic shapes. Overall, the work is done with care and thoroughness. I am a bit hesitant in recommending acceptance of this work. The reason is I am not sure how easily this method can be implemented by those in the field. While it is true that microfluidic methods to produce emulsions have become quite popular in the field, it is still limited to a small number of research groups that have the specific facilities and skill sets. How scalable would this technique be? It is true that recent advances in scale-up of microfluidics have been made but they are mostly limited to single emulsion scale-up.

We thank the reviewer for voicing the concerns. We would like to clarify that at present, no other method enables control over surface coverage and composition and we do not directly see a technique which could achieve this, nor has the possibility to obtain quantitative control been achieved.

The comment about microfluidic techniques requiring specific facilities is a bit surprising to us. All the methods and tools are readily available, and several commercial companies exist which could easily make the chips used in this work. All other equipment (syringe pumps and microscopes) is standard.

The question concerning scalability for double emulsions has been addressed indeed only very recently by different groups. The key idea is scale up of microfluidic emulsification using step emulsification followed by reinjection [1,3]. The step emulsification takes away many of the problems of parallelization and nozzle clogging. The reinjection is simple and just requires two devices to be connected, allowing efficient production of double emulsions. A single chip of the radial design presented in ref. 3. enables one to produce several liters of the monodisperse emulsions per day. The scale up goes beyond the scope of the present paper, but will be addressed in the near future. We have clarified the ease of scale up in the revised manuscript, and added 2 additional references.

“Scale up of microfluidic methods has gained momentum, even for double emulsions [3] enabling production of litre quantities per day for monodisperse emulsions. The key idea is to utilize step emulsification devices in tandem to enable re-injection of pre-formed droplets. The step emulsification takes away many of the problems of parallelisation and nozzle clogging [2]. The re-injection is simple and just requires two devices to be connected, allowing efficient production of double emulsions [1].”

- [1] Eggersdorfer, M. L., Zheng, W., Nawar, S., Mercandetti, C., Ofner, A., Leibacher, I., ... & Weitz, D. A. (2017). Tandem emulsification for high-throughput production of double emulsions. *Lab on a Chip*, 17(5), 936-942.
- [2] Stolovicki, E., Ziblat, R., & Weitz, D. A. (2018). Throughput enhancement of parallel step emulsifier devices by shear-free and efficient nozzle clearance. *Lab on a Chip*, 18(1), 132-138.
- [3] Ofner, A., Moore, D. G., Rühls, P. A., Schwendimann, P., Eggersdorfer, M., Amstad, E., ... & Studart, A. R. (2017). High-Throughput Step Emulsification for the Production of Functional Materials Using a Glass Microfluidic Device. *Macromolecular Chemistry and Physics*, 218(2), 1600472.

For conventional Pickering emulsions, the surface concentration is not controlled, but a consequence of a process of limited coalescence [4]. The work presented here demonstrates how a genuine transient double and then a simplified double emulsion can be used to control the surface coverage AND composition, for a wide range of materials, indeed going from liposomes, over polymerosomes to pickering emulsions, but also mixed composite interfaces with composition control (which is not possible with other technologies). We tried to underline the generic nature of the concept, emphasize the quantification and demonstrate some unique features. As a demonstrator of the versatility we showed a droplet containing both hydrophobic and hydrophylic particles, and we do not see any technology which could deliver this. The simplified double emulsion (slow dissolution) opens up this method to a very wide range of chemistries. We hope to have clarified and emphasized this in the revised version of the manuscript.

[4] Arditty, S., Whitby, C. P., Binks, B. P., Schmitt, V., & Leal-Calderon, F. (2003). Some general features of limited coalescence in solid-stabilized emulsions. *The European Physical Journal E*, 11(3), 273-281.

We indeed refrained from a detailed characterization of the specific monolayers used as the goal of the manuscript was to show the precision obtained over coverage; we used particles with near hard sphere behavior. Clearly fundamental studies of the link between layer properties (isotherms, interfacial rheology) and droplet deformation and stability are on the agenda, but go beyond the scope of the present work.

I also have a few additional questions:

1. I am a bit confused by the how the inner phase evolves over time as its composition changes from Point I to Point D in Figure 1. Is there any phase separation that is observed during the evolution of the droplet morphology as the middle phase dissolves into the inner phase?

The principle used is the same as the one in liquid-liquid extraction, meaning that the middle phase solvent is extracted to the inside or the outside. The system remains two phasic until complete dissolution of the middle phase. In the simplified method this is of course not the case. We have added a comment for clarification to the text at this point.

“The principle is similar to liquid-liquid extraction, but with complete dissolution of the middle phase leaving the insoluble material on the interface.”

2. Could this method be used to produce sub-micron/nano droplets? These are the droplets that are very difficult to make and control using the conventional methods of emulsification.

Currently the size ranges are the ones limited by the microfluidic regime hence in the micrometric range, but possibly with dissolving droplets size reduction could be achieved.

3. Figure 1S(g) shows that some droplets remains as single droplets whereas some undergo arrested coalescence. Isn't this showing that not all droplets have uniform coverage and there are some heterogeneity in the particle coverage as also shown in the non-negligible scatter and error range in Figure 3(a)?

We thank the reviewer for pointing this out, we should have been more clear with respect to this point. The specific experiment is right at the minimal surface coverage for stability. The coverage was homogeneous from one drop to another, but we see accidental coalescence (which is not a fully deterministic process). The very fact that the droplet remains nonspherical after coalescence confirms the near maximum packing of the surface coverage. Detailed

analysis of coalescence lies beyond the scope of this work, but it is clear that with the method developed here the detailed role of surface coverage and subsequent charging, steric hinderance or interfacial rheology on coalescence can be uniquely investigated. We have added a clarification to the text on page 10.

“The specific experiment shown here was at the minimal surface coverage where we expect stability against coalescence. The coverage was homogeneous from one drop to another, but exactly at 12 mg/ml we observe accidental coalescence (which is not a fully deterministic process). The coalesced droplets remained anisotropic in shape, confirming the near maximum packing of the surface coverage just before coalescence.”

Replies to the comments of reviewer 3.

The paper “Designer liquid-liquid interfaces made from out-of-equilibrium double emulsions” demonstrates a new method for producing particle stabilized droplets with the following advantages: controlled surface coverage; uniform surface coverage over a population of droplets; monodisperse droplets made in microfluidics. The first two points in particular represent an outstanding challenge in the field of emulsification and formulations, that is addressed here for the first time. The proposed approach is very attractive and I have no doubt that it will be taken up by many researchers in the field.

The idea presented is simple and powerful. Droplets are formed from two immiscible phases as usual. The material that is to form the coating on the droplet surface is suspended in an auxiliary phase, which is miscible with either the continuous or the dispersed phase. A double emulsion is formed, so that the a layer of the auxiliary phase initially separates the two immiscible phases. The auxiliary phase diffuses in the phase in which it is miscible, and the suspended material is left as a coating on the droplets. The parameters that determine the surface coverage (flow rate of auxiliary phase, etc) are all controlled in practice, unlike in existing methods.

I really like the concept presented in this paper and the results shown are very compelling. My questions and comments are below.

COMMENTS

1. I have a question as to the generality of this approach, since it requires identifying an auxiliary phase with suitable miscibility, and also in which the material (colloids in this case, but could be proteins or polymers) is stable in the colloidal sense. Could the authors reassure us in this respect?

The work presented here shows that true binary or ternary phases are not required, just slow dissolution seems to work. Clearly a lot is to be explored. The middle, auxiliary phase will always be intermediate between the two phases in terms of polarity, and for the surface active material to be surface active that will often be the case as well. We have added a reassuring statement to the text, pointing out the degrees of freedom offered by the proposed method, on page 16/17 where we also address the comment 2&3 of reviewer 1.

“The main concern for the chemicals is that the middle or auxiliary phase should enable dispersion or dissolution of the material which one wants to deposit onto the interface. This middle, phase will always be intermediate between the two other phases in terms of polarity, and for the surface active material to be surface active that will often be the case as well. Moreover, we have found that the condition of transient immiscibility could be relaxed to one of slow dissolution.”

2. I would like to encourage the authors to take some time to improve the clarity of the manuscript, which is a bit difficult to follow at the moment. The first three sentences in the abstract are obscure to a non-expert reader. The Introduction is unclear: thermodynamics, kinetics, and other fundamental concepts are mentioned superficially but the details relevant to the system of interest remain vague (e.g. what is meant with “The immiscibility can come from the underlying phase diagram or be kinetic in nature”?). The Introduction is also a bit repetitive, especially on the discussion of microfluidics approaches.

Thanking you for pointing this out, we have undertaken an effort to clarify the manuscript and removed vague and repetitive sections.

3. What are the shaded areas in the graph in Figure 3a?

We have clarified that these represent the uncertainty limits (propagation of the error of particle counting explained in the text).

“... with the shaded areas a conservative error obtained from the image analysis as explained in the text the image analysis as explained in the text.”

“The error was estimated by evaluating the sensitivity of the method by slight changes of the circle radius, due to particles lying on the edge of the circle.”

MINOR POINTS

4. Non-equilibrium double emulsion (as in the title) or transient double emulsion (as in the paper)? I personally find “transient” to more clearly convey the mechanism behind this new approach

Agreed. We have changed the title according to your suggestion.

5. Figure 4b left me wondering if in this image the particles are in a monolayer at the interface, in a multi layer at the interface, or in the miscible liquid layer before it dissolves. Can this be made clearer in the caption?

We do not have confocal contrast for this system so we cannot be very quantitative. Based on the mass balance we would expect a thicker layer. The image was taken after the miscible liquid layer had dissolved. We have added this to the caption.

“... this image was taken after the miscible liquid had dissolved and a multilayer was formed.”

REVIEWERS' COMMENTS:

Reviewer #1 (Remarks to the Author):

Thank you for the careful attention to the reviews. I am satisfied with the authors' responses and the changes they have made to the manuscript. I recommend the revised work be published.

Reviewer #2 (Remarks to the Author):

The authors have given thoughtful response to the comments raised by the reviewers. Overall, I am now of opinion that the manuscript deserves publication in Nature Communications as it describes a unique method of controlling Pickering emulsions.

Reviewer #3 (Remarks to the Author):

The authors have addressed the comments by the reviewers to my satisfaction and warmly recommend for publication.